

# Palmitic acid induces neurotoxicity and gliatoxicity in SH-SY5Y human neuroblastoma and T98G human glioblastoma cells

Yee-Wen Ng and  Yee-How Say

Department of Biomedical Science, Faculty of Science, Universiti Tunku Abdul Rahman (UTAR) Kampar Campus, Kampar, Perak, Malaysia

Corresponding author
Yee-How Say, sayyh@utar.edu.my, howsyh@gmail.com

## ABSTRACT

**Background**. Obesity-related central nervous system (CNS) pathologies like neuroinflammation and reactive gliosis are associated with high-fat diet (HFD) related elevation of saturated fatty acids like palmitic acid (PA) in neurons and astrocytes of the brain.

**Methods**.  Human neuroblastoma cells SH-SY5Y (as a neuronal model) and human glioblastoma cells T98G (as an astrocytic model), were treated with 100–500 μM PA, oleic acid (OA) or lauric acid (LA) for 24 h or 48 h, and their cell viability was assessed by 3-(4,5-dimethylthiazol-2-yl)-2,5-diphenyltetrazolium bromide (MTT) assay. The effects of stable overexpression of γ-synuclein (γ-syn), a neuronal protein recently recognized as a novel regulator of lipid handling in adipocytes, and transient overexpression of Parkinson's disease (PD) α-synuclein [α-syn; wild-type (wt) and its pathogenic mutants A53T, A30P and E46K] in SH-SY5Y and T98G cells, were also evaluated. The effects of co-treatment of PA with paraquat (PQ), a Parkinsonian pesticide, and leptin, a hormone involved in the brain-adipose axis, were also assessed. Cell death mode and cell cycle were analyzed by Annexin V/PI flow cytometry. Reactive oxygen species (ROS) level was determined using 2′,7′-dichlorofluorescien diacetate (DCFH-DA) assay and lipid peroxidation level was determined using thiobarbituric acid reactive substances (TBARS) assay.

**Results**. MTT assay revealed dose- and time-dependent PA cytotoxicity on SH-SY5Y and T98G cells, but not OA and LA. The cytotoxicity was significantly lower in SH-SY5Y-γ-syn cells, while transient overexpression of wt α-syn or its PD mutants (A30P and E46K, but not A53T) modestly (but still significantly) rescued the cytotoxicity of PA in SH-SY5Y and T98G cells. Co-treatment of increasing concentrations of PQ exacerbated PA's neurotoxicity. Pre-treatment of leptin, an anti-apoptotic adipokine, did not successfully rescue SH-SY5Y cells from PA-induced cytotoxicity—suggesting a mechanism of PA-induced leptin resistance. Annexin V/PI flow cytometry analysis revealed PA-induced increase in percentages of cells in annexin V-positive/PI-negative quadrant (early apoptosis) and subG_0-G_1 fraction, accompanied by a decrease in G_2-M phase cells. The PA-induced ROS production and lipid peroxidation was at greater extent in T98G as compared to that in SH-SY5Y.

**Discussion**. In conclusion, PA induces apoptosis by increasing oxidative stress in neurons and astrocytes. Taken together, the results suggest that HFD may cause

neuronal and astrocytic damage, which indirectly proposes that CNS pathologies involving neuroinflammation and reactive gliosis could be prevented via the diet regimen.

## INTRODUCTION

Obesity is now a global health issue that presents a major risk for serious diet-related noncommunicable diseases including diabetes mellitus, cardiovascular diseases, hypertension, stroke, and some cancers (*World Health Organization, 2018*). High-fat diet (HFD) rich in saturated fatty acids (SFAs) has long been recognised to contribute to many adverse metabolic health issues. In the past two decades, it was found that HFD-induced obesity has been associated with neuroinflammation and reactive gliosis (*Dorfman & Thaler, 2015*), leading to CNS pathologies such as neurodegenerative diseases (*Guillemot-Legris & Muccioli, 2017*). A case-control study demonstrated a link between the intake of fat from animal sources rich in SFA and PD (*Logroscino et al., 1996*). Several prospective studies also showed association of body mass index (BMI) with increased risk of Alzheimer's disease (*Gustafson et al., 2003*) and PD (*Hu et al., 2006*); and on the contrary, lower BMI is associated with lower risk of PD (*Sääksjärvi et al., 2014*). Furthermore, the first study in humans using positron emission tomography with [$^{11}$C]-palmitate and [$^{18}$F]fluoro-6-thia-heptadecanoic acid showed increased fatty acid (FA) uptake and accumulation in the brain of obese subjects with metabolic syndrome (*Karmi et al., 2010*). This suggests that FAs are able to cross the blood–brain-barrier and are able to be taken up by brain cells. In fact, peripheral FAs were shown to have relationship with central FAs, as reported by a study done on human whole blood and cerebrospinal fluid (*Guest et al., 2013*).

PA, a long chain 16:0 SFA, is the most common FA found in animals and plants such as palm oil, palm kernel oil, butter, cheese, milk and meat (*Gunstone, Harwood & Dijkstra, 2010*). Despite its crucial biological functions such as energy storage, acting as a signalling molecule and maintaining integrity of cellular membranes (*Gunstone, Harwood & Dijkstra, 2010*), PA has been found to be increased in diseased brains. Particularly, PA level appeared to be increased in the frontal cortex lipid rafts in PD (*Fabelo et al., 2011*); and in parietal cortex in AD (*Fraser, Tayler & Love, 2010*). Furthermore, PA was found to cause lipotoxicity to several cell lines *in vitro*. For instance, PA triggered the release of tumour necrosis factor-α and interleukin-6, activating inflammatory signalling in astrocytes (*Gupta et al., 2012*). PA also induced apoptosis in human hepatoma cell line, HepG2 (*Zhang et al., 2004*), neural progenitor cells (*Park et al., 2011*) and neuronal cell line, SH-SY5Y (*Hsiao et al., 2014*).

Other than neurons, the brain also comprises and depends on surrounding non-neuronal cells such as glial, epithelial cells, pericytes and endothelia, for them to function correctly (*Freire-Regatillo et al., 2017*). Glial cells were once thought of to serve as a supportive system
for neurons. Now they are found to possess modulatory, trophic and immune functions in the brain (*Gupta et al., 2012*). Astrocyte is a type of glial cell, which is the most plentiful and varied non-neuronal cell in the CNS (*Argente-arizón et al., 2015*). Of note, marked astrogliosis was observed in the hypothalamus of HFD-induced or genetically obese mice (*Buckman et al., 2013*), suggesting that astrocyte plays a role in reactive gliosis leading to CNS pathologies.

A family of neuronal proteins that are implicated in both obesity and CN pathologies is the synucleins, i.e., α- and γ-synucleins (α-syn, γ-syn). They are small, soluble, highly conserved proteins, predominantly expressed in the neurons. α-syn is a major component of Lewy bodies and Lewy neurites appearing in the postmortem brain of PD and other synucleinopathies (*Spillantini & Goedert, 2000*). Genetic mutations in α-syn, including point mutations-A53T, A30P and E46K, have been linked to familial PD (*Polymeropoulos et al., 1997*). Neuronal expression of either human wild-type (wt) or PD-related mutant α-syn induces neurodegeneration associated with pathological accumulations of α-syn. Previous studies also revealed that α-syn-containing inclusion bodies were present in astrocytes of sporadic PD (*Wakabayashi et al., 2000*; *Braak, Sastre & Del Tredici, 2007*) and overexpression of wt α-syn in U373 astrocytoma cells induces of astroglial apoptotic cell death (*Stefanova et al., 2001*). Moreover α-syn and γ-syn were found to affect lipid uptake and metabolism in brain and adipocytes (*Golovko et al., 2005*; *Millership et al., 2013*; *Hsiao, Halliday & Kim, 2017*).

Given the involvement of FAs particularly PA in CNS pathologies and the possible role of α-syn and γ-syn in modulating the effects of FAs, therefore, the objective of this study was to first evaluate the effects of PA, oleic acid (OA; long chain FA with lipid number of C18:1 *cis-9* and a major constituent in plant oil such as olive oil, almond oil, pecan oil and canola oil) and lauric acid (LA; medium chain 12:0 SFA which comprises about 50% of FA content in coconut oil, coconut milk, laurel oil and palm kernel oil) on the viability of human neuroblastoma SH-SY5Y and human glioblastoma T98G cell lines. SH-SY5Y cells were selected for the experiments as they have been widely used as a cell model of dopaminergic neurons for PD research (*Xie, Hu & Li, 2010*), while T98G cells were selected due to its biological resemblance with primary astrocytes and its broad use in research as an astrocyte cell model (*Avila Rodriguez et al., 2014*; *Cabezas et al., 2015*; *Avila-Rodriguez et al., 2016*). The effects of stable overexpression of γ-syn in SH-SY5Y and transient overexpression of α-syn (wt and PD mutants A53T, A30P and E46K) in SH-SY5Y and T98G cells were also evaluated. We found that PA is neurotoxic and gliatoxic to SH-SY5Y and T98G cells, respectively. To investigate the potential synergistic effect of environmental factors for dopaminergic neurotoxicity, SH-SY5Y cells were co-treated with PA (to mimic HFD exposure), and increasing concentrations of paraquat (PQ), a herbicide that is implicated in the development of PD (*Pezzoli & Cereda, 2013*). Since leptin, a hormone that is involved in the brain-adipose axis, has been shown to have neuroprotective effect in SH-SY5Y cells (*Russo et al., 2004*; *Lu et al., 2006*; *Weng et al., 2007*), we also investigated whether leptin pre-treatment could rescue SH-SY5Y cells from PA neurotoxicity. The mode of cell death induction by PA in SH-SY5Y and T98G was investigated using Annexin V/PI staining followed by flow cytometry analysis. Lastly, to
attribute whether apoptotic cell death is caused by oxidative stress, intracellular ROS and extent of lipid peroxidation (TBARS level) were assessed.

## MATERIALS AND METHODS

### Cell culture, transfections and treatments

SH-SY5Y (ATCC® CRL-2266[TM]) and T98G (ATCC® CRL-1690[TM]), obtained from the American Type Culture Collection (ATCC), were maintained in Eagle's Minimum Essential Medium (MEM) (Corning, NY, USA) and Dulbecco's Modified Eagle's Medium (DMEM) (Corning, NY, USA), respectively, supplemented with 10% (v/v) fetal bovine serum (Sigma-Aldrich, St. Louis, MO, USA) and 1% (v/v) penicillin–streptomycin (Nacalai Tesque, Osaka, Japan) at 37 °C and 5% $CO_2$ in air. All cell lines have been checked to ensure they are free of contamination and have been used from young stock (less than seven passages). SH-SY5Y overexpressing γ-Syn (SH-SY5Y-γ) was established by stable transfection of SH-SY5Y cells with plasmid pOTB7 carrying full length human $\gamma$-syn (accession number: BC014098) cDNA (clone ID: 4546444; obtained from Addgene, Cambridge, MA, USA). Transfection was performed by electroporation using ECM® 830 ElectroSquarePorator[TM] (BTX Harvard Apparatus, Holliston, MA, USA) and cells were cultured in complete growth medium for 72 h prior to 14 days of antibiotic selection with 1 mg/ml of G418 sulfate (A.G. Scientific Inc., San Diego, CA, USA). Following that, γ-syn protein expression was assessed by Western blot using anti-γ-synuclein primary antibody, clone EP1539Y (Millipore, Burlington, MA, USA) prepared in 1:5,000 dilution and secondary HRP-conjugated anti-rabbit IgG antibody (Sigma Aldrich, St. Louis, MO, USA) prepared in 1:10,000 dilution (data shown in raw data file link as stated in 'Data Availability').

Transient transfection of α-syn (wt, PD mutants A53T, A30P or E46K) cDNAs (*Choong & Say, 2011*) or γ-Syn cDNA was performed in 96-well plates using TurboFect[TM] Transfection Reagent (Cat. No.: R0531; Thermo Fisher Scientific, CA, USA) according to the manufacturer's protocol.

The FA-bovine serum albumin complex was prepared according to the protocols established by *Hsiao et al. (2014)*, albeit with slight modification. Briefly, 20 mM of FA was prepared in 0.01 N sodium hydroxide in a dry bath of 80 °C. MEM or DMEM with 1% BSA was added to different volumes of FA stock solution to reach final concentrations of 100–500 μM. The mixtures were incubated in the 37 °C water bath for 30 min before being treatment of cell lines. The cell lines were first treated with increasing concentrations (0, 100, 200, 300, 400 and 500 μM) of PA (Merck, Kenilworth, NJ, USA), OA (Nacalai Tesque, Japan) or LA (Merck, Kenilworth, NJ, USA) to determine the cytotoxicity of various FAs. The untreated (0 μM) consisted of MEM or DMEM with 1% BSA. After that, different concentrations were used for different assays as described in the following sections.

### MTT cell viability assay

SH-SY5Y/SH-SY5Y-γ ($1.2 \times 10^4$ cells/well) or T98G ($8 \times 10^3$ cells/well) cells were seeded into 96-well plates, treated with different treatment paradigms and incubated for 24 or

48 h. Medium with vehicle (BSA) was used as blank. Then, 20 µl of 5 mg/ml 3-(4,5-dimethylthiazol-2-yl)-2,5-diphenyltetrazolium bromide (MTT; Bio Basic Inc., Ontario, Canada) prepared in phosphate buffer saline (PBS), was added to each well and the plate was incubated at 37 °C for 4 h. The medium in each well was then discarded and 100 µl of dimethyl sulfoxide (DMSO) was added into each well. The plate was incubated at 37 °C for 30 min prior to absorbance reading at 560 nm using FLUOstar® Omega Microplate Reader (BMG LABTECH, Ortenberg, Germany).

## Cell death mode and cell cycle analyses by flow cytometry

The cell death mode of SH-SY5Y and T98G cell lines induced by PA was determined using flow cytometry. The cells were stained with annexin V and PI using Alexa Fluor® 488 Annexin V/Dead Cell Apoptosis Kit (Cat. No.: V13241; Thermo Fisher Scientific, Waltham, MA, USA). The assay was performed according to the manufacturer's protocol. Briefly, the untreated and treated cells were trypsinised, collected and washed with PBS. Then, the cells were pelleted at 800 g for 5 min. The cells were resuspended at the density of $1 \times 10^6$ cells/ml with $1 \times$ annexin-binding buffer. To every 100 µl of cell suspension, 5 µl of Alexa Fluor® 488 annexin V and 1 µl of 100 µg/ml PI (prepared by adding 5 µl of 1 mg/ml stock to 45 µl of $1 \times$ annexin-binding buffer) were added. The cells were incubated at room temperature for 15 min. After that, 400 µl of $1 \times$ annexin-binding buffer was added and the cells were analysed immediately using Attune$^{TM}$ Nxt Flow Cytometer (Thermo Fisher Scientific, Waltham, MA, USA). Alexa Fluor® 488 annexin V and PI fluorescence emission was measured using the 530/30 nm (BL1) and 695/40 nm (BL3) emission filters, respectively, with excitation at 488 nm. A total of 10,000 events per sample was recorded. Data collected were analysed with Attune$^{TM}$ Nxt Software. Compensation was set up using unstained cells, cells stained with Alexa Fluor® 488 annexin V only and cells stained with PI only. Negative gating strategy involving forward scatter *vs.* side scatter plot was to eliminate cell debris.

Cell cycle analysis was performed according to *Henry, Hollville & Martin (2013)* with slight modification. Briefly, untreated and treated cells were harvested and washed with PBS. The cells were then fixed with 1 ml 70% ethanol at −20 °C for 1 h. After that, the cells were centrifuged at 2,500 g for 5 min and the supernatant was discarded. The pellet was resuspended with 1 ml phosphate-citrate wash buffer (200 mM $Na_2PO_4$ (Merck, USA), 100 mM citric acid (Merck, Kenilworth, NJ, USA)) followed by centrifugation at 2,500 g for 5 min. The supernatant was discarded and the cell pellet was resuspended in PI staining solution containing 10 µg/ml PI and 100 µg/ml RNase A prepared in PBS. The cells were analysed using the Attune$^{TM}$ Nxt Flow Cytometer with the 695/40 nm (BL3) emission filter. The data collected were analysed using Attune$^{TM}$ Nxt Software.

## Intracellular ROS quantification by DCFH-DA assay

Intracellular ROS level was detected using the fluorescent probe DCFH-DA (Sigma-Aldrich, St. Louis, MO, USA). SH-SY5Y or T98G cell line was seeded with phenol red-free complete medium in black 96-well plates overnight, and 25 µM DCFH-DA prepared in complete medium was added into each well. The cells were incubated at 37 °C for 45 min before the

medium was removed from each well. Next, 100 µl of FA treatment was added and the plate was incubated at 37 °C for 6 h. The fluorescent signal was read at Ex485nm/Em535nm using FLUOstar® Omega Microplate Reader (BMG LABTECH, Ortenberg, Germany). Untreated, unstained cells were used as blank.

## Quantification of lipid peroxidation using TBARS assay

Parameter$^{TM}$ TBARS Assay kit (R&D Systems, Minneapolis, MN, USA) was used to quantify lipid peroxidation by measuring TBARS levels, as an indicator of oxidative stress in cells. The assay was performed in SH-SY5Y and T98G cells after PA treatment, according to the manufacturer's protocol. Briefly, cell lysate was prepared by first collecting the cells and washing the cells with cold PBS. Then, the cells were resuspended in deionised water at the density of $1 \times 10^6$ cells/ml. The cell suspension was subjected to a total of 3 cycles of 10-s sonication and then freeze/thaw at $\leq -20$ °C. Then, 300 µl of the cell lysate was subjected to acid treatment with 300 µl of TBARS Acid Reagent provided in the kit. After 15 min of incubation at room temperature, the mixture was centrifuged at $\geq 12,000$ g for 4 min and the supernatant was retained. Next, 150 µl of standards and samples were added into each well of the microplate and 75 µl of TBA Reagent was added. The optical density of each well was pre-read at 532 nm using FLUOstar® Omega Microplate Reader (BMG LABTECH, Ortenberg, Germany). Then, the microplate was covered with the adhesive strip provided and was incubated at 45–50 °C for 2–3 h. After incubation, the optical density was read at 523 nm. The pre-reading was subtracted from the final reading to correct for the samples contribution to the final absorption at 532 nm. The results were calculated according to the manufacturer's instruction. A linear standard curve was plotted and the concentrations of samples were read from the standard curve and were multiplied by the dilution factor 2.

## Statistical analysis

Quantitative data were presented as mean ± standard error of mean (S.E.M.) from triplicates of three independent experiments, unless otherwise stated. Statistical analysis was performed using Statistical Package for the Social Sciences (SPSS) software (version 23.0) (SPSS Inc., Chicago, IL, USA). All results were subjected to paired-sample $t$-test. A $p$-value of $<0.05$ was considered as statistically significant.

# RESULTS

## PA, but not OA and LA, is neurotoxic and gliatoxic

Generally, PA induced cytotoxicity towards SH-SY5Y and SH-SY5Y-γ in a time- and dose-dependent manner at concentrations above 300 µM at 24 h, and at concentrations above 200 µM at 48 h. Figures 1A and 1B show the effect of PA treatment on the cell lines for 24 and 48 h. At 24 h, the percentage of viable cells decreased dramatically at PA concentrations $\geq 300$ µM, while at 48 h, the cell viability decreased drastically at PA concentrations $\geq 200$ µM. Except at 500 µM, the viability of SH-SY5Y-γ cells was significantly higher than SH-SY5Y cells at concentrations $\geq 300$ µM (24 h) and at concentrations $\geq 200$ µM (48 h). The concentrations of 50% cytotoxicity (LC$_{50}$) for SH-SY5Y and SH-SY5Y-γ were 420 and 440 µM, respectively, at 24 h, and 320 and 380 µM, respectively, at 48 h. At lower

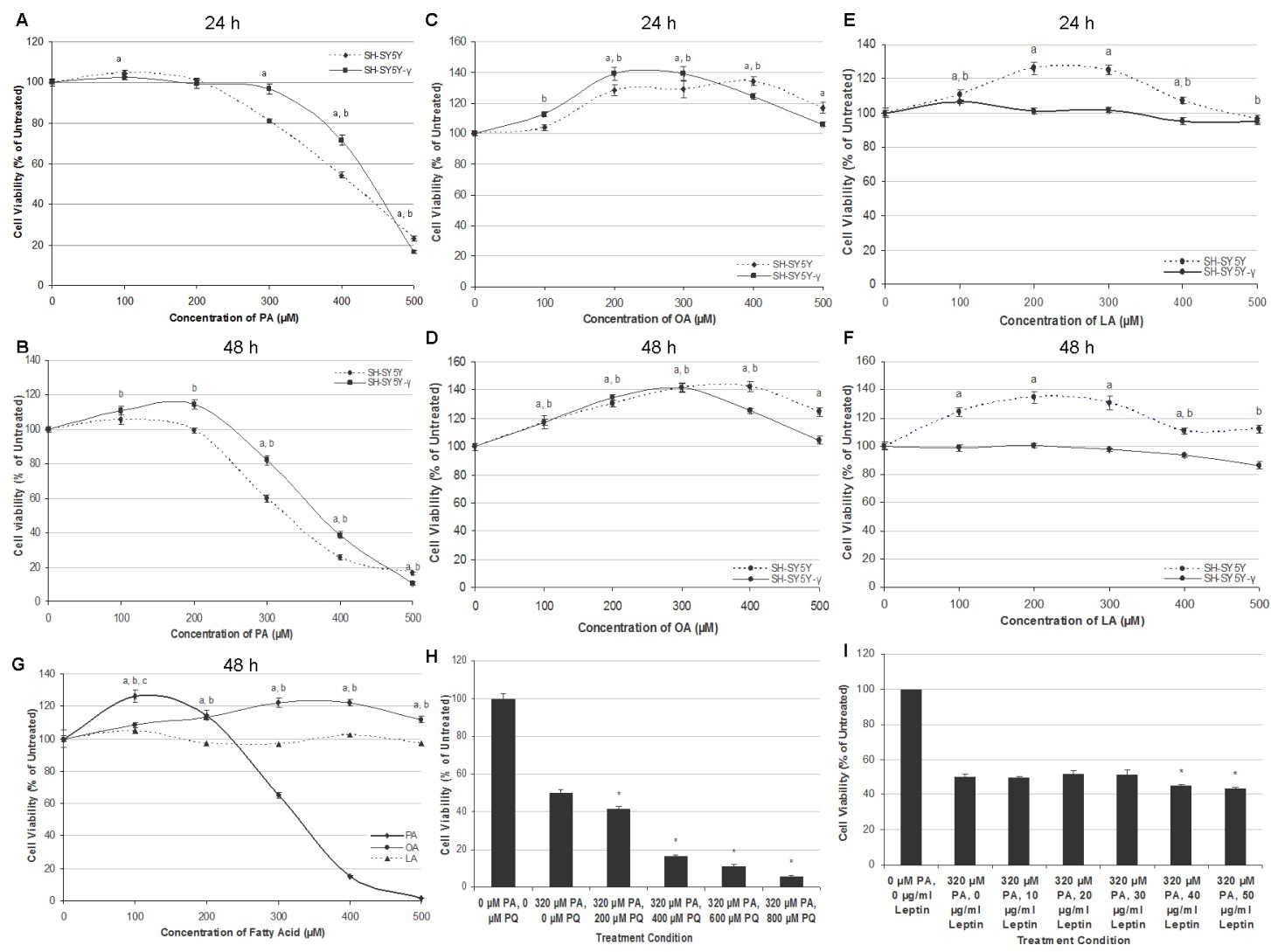

**Figure 1** **PA, but not OA or LA, is neurotoxic and gliatoxic to SH-SY5Y and T98G cells, and the effects are ameliorated by γ-syn overexpression and exacerbated by PQ treatment.** SH-SY5Y and SH-SY5Y-γ cells were treated with increasing concentrations of PA (A, B), OA (C, D) and LA (E, F) for 24 h (A, C, E) or 48 h (B, D, F). Data represent mean ± S.E.M. of three independent experiments; a and b represent $p < 0.05$ as compared to the untreated for SH-SY5Y and SH-SY5Y-γ, respectively. (G) Effects of fatty acid treatments in T98G cell line. T98G cells were treated with increasing concentrations (0, 100, 200, 300, 400 and 500 μM) of PA, OA and LA for 48 h; a, b and c represent $p < 0.05$ as compared to the untreated for PA, OA and LA, respectively. (H) SH-SY5Y cells were co-treated with 300 μM PA and with increasing concentrations (0, 200, 400, 600, and 800 μM) of PQ for 48 h. MTT assay was then performed. Data represent mean ± S.E.M. of three independent experiments; * represents $p < 0.05$ as compared to 300 μM of PA treatment. (I) SH-SY5Y cells were pre-treated with increasing concentrations (0, 10, 20, 30, 40 and 50 μg/ml) of leptin for 6 h followed by 300 μM PA treatment for 48 h. MTT assay was then performed. Data represent mean ± S.E.M. of three independent experiments; * represents $p < 0.05$ as compared to untreated.

concentrations, PA did not cause cytotoxicity but promoted the viability of SH-SY5Y-γ instead at 48 h.

The treatment of OA for 24 and 48 h generally increased the cell viability in both SH-SY5Y and SH-SY5Y-γ, as shown in Figs. 1C and 1D. The cell viability increased as a function of OA concentration and reached the peak of ∼140% at 400 μM and 300 μM

OA for SH-SY5Y and SH-SY5Y-γ, respectively. For the treatment of LA, the cell viability of SH-SY5Y, but not that of SH-SY5Y-γ, was generally increased, as shown in Figs. 1E and 1F. The results indicate that LA promoted the viability of SH-SY5Y but had no effect on SH-SY5Y-γ at concentrations <300 μM and exerted significant decrease in cell viability (8–18%) at ≥400 μM at 48 h.

Similar with SH-SY5Y cells, PA at lower concentrations (<200 μM), increased the viability of astrocytic T98G cells, and while at higher concentrations (≥200 μM), reduced the cell viability significantly, in a dose-dependent manner (Fig. 1G). The $LC_{50}$ of PA in T98G cells was 320 μM. Like in SH-SY5Y cells, OA also increased the cell viability of T98G at concentrations ≥200 μM, while LA did not affect the cell viability significantly (Fig. 1G).

## Co-treatment of PQ exacerbates neurotoxicity of PA, but leptin did not ameliorate the neurotoxicity of PA

Co-treatment of PQ exacerbates the neurotoxicity of PA in a dose-dependent manner, as illustrated in Fig. 1H. At 200 μM PQ, the cell viability decreased to $41.6 \pm 1.2\%$ (Fig. 1H). As compared to PQ treatment alone, the $LC_{50}$ was determined to be 380 μM (data shown in raw data file link as stated in the 'Data Availability' section). Interestingly, pre-treatment of increasing concentrations of leptin for 6 h not only did not rescue the cells from PA neurotoxicity, but further exacerbates PA neurotoxicity at concentrations ≥40 μg/ml (Fig. 1I).

## Transient transfection of α-syn and its PD mutants led to modest rescue from the neurotoxicity of PA

To investigate the gene-environment interaction in affecting neurotoxicity and gliatoxicity, SH-SY5Y and T98G cells were transiently transfected with wt, A30P, E46K or A30P α-syn for 48 h before further treated with $LC_{50}$ of PA (320 μM for both SH-SY5Y and T98G cells). Instead of exacerbating the cytotoxic effect of PA, transient overexpression of wt α-syn or its PD mutants (A30P and E46K, but not A53T) significantly rescued SH-SY5Y cells from the toxicity of PA, by increasing their viability to up to 10% (Fig. 2A). The similar phenomenon could also be observed in T98G cells, where transient overexpression of A30P and E46K significantly increased the cell viability to around 10%, compared with the untransfected cells treated with PA (Fig. 2B). Furthermore, transient overexpression of wt, A30P or E46K α-syn also increased the viability of SH-SY5Y cells (but not T98G) to up to 20% when subjected to OA treatment (Fig. 2A).

Since the discrepancy in the cytotoxic effect of α-syn overexpression might be due to the transfection method used (transient in this study *vs.* stable in *Stefanova et al., 2001*), we investigated whether the neurotoxicity results of PA in stable SH-SY5Y-γ cells could be replicated by transient transfection of γ-syn in SH-SY5Y and T98G cells. Indeed, transient transfection of γ-syn did not significantly increase the cell viability of SH-SY5Y cells when subjected to PA treatment (compared with results in Fig. 1B). In short, transient transfection of α-syn or γ-syn will have different effects in SH-SY5Y or T98 cells compared with stable transfection.

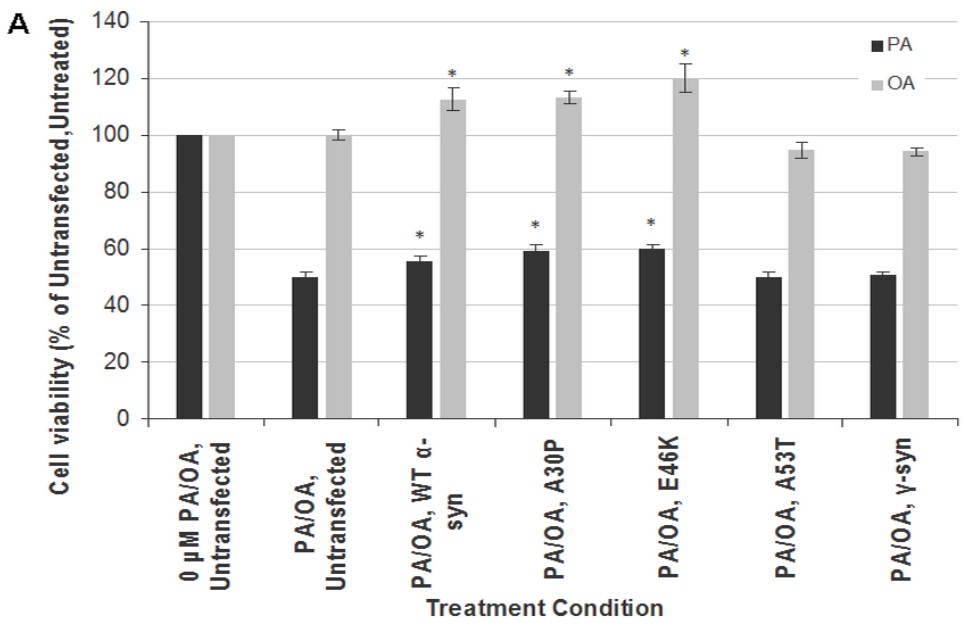

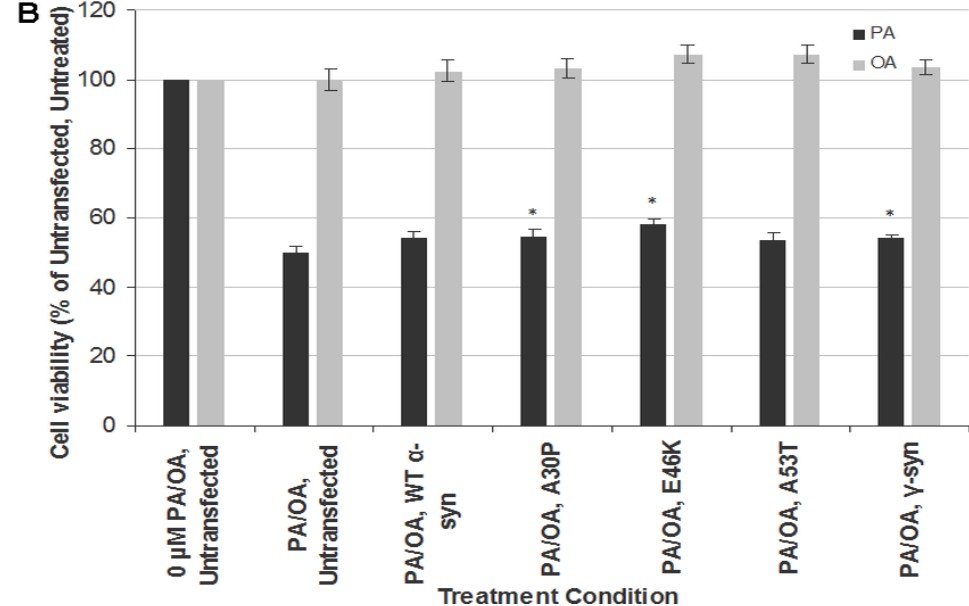

**Figure 2** **Effects of PA and OA on α-syn or γ-syn transient-transfected SH-SY5Y and T98G cells.** SH-SY5Y and T98G cells were transfected with wild type (WT), different mutants of α-syn (A30P, E46K and A53T) and γ-syn for 48 h followed by the treatment of 300 μM of PA or 100 μM OA for 48 h. Then, MTT assay was performed. (A) SH-SY5Y. (B) T98G. Data represent mean ± S.E.M. of three independent experiments; * represents $p < 0.05$ as compared to the untransfected, untreated.

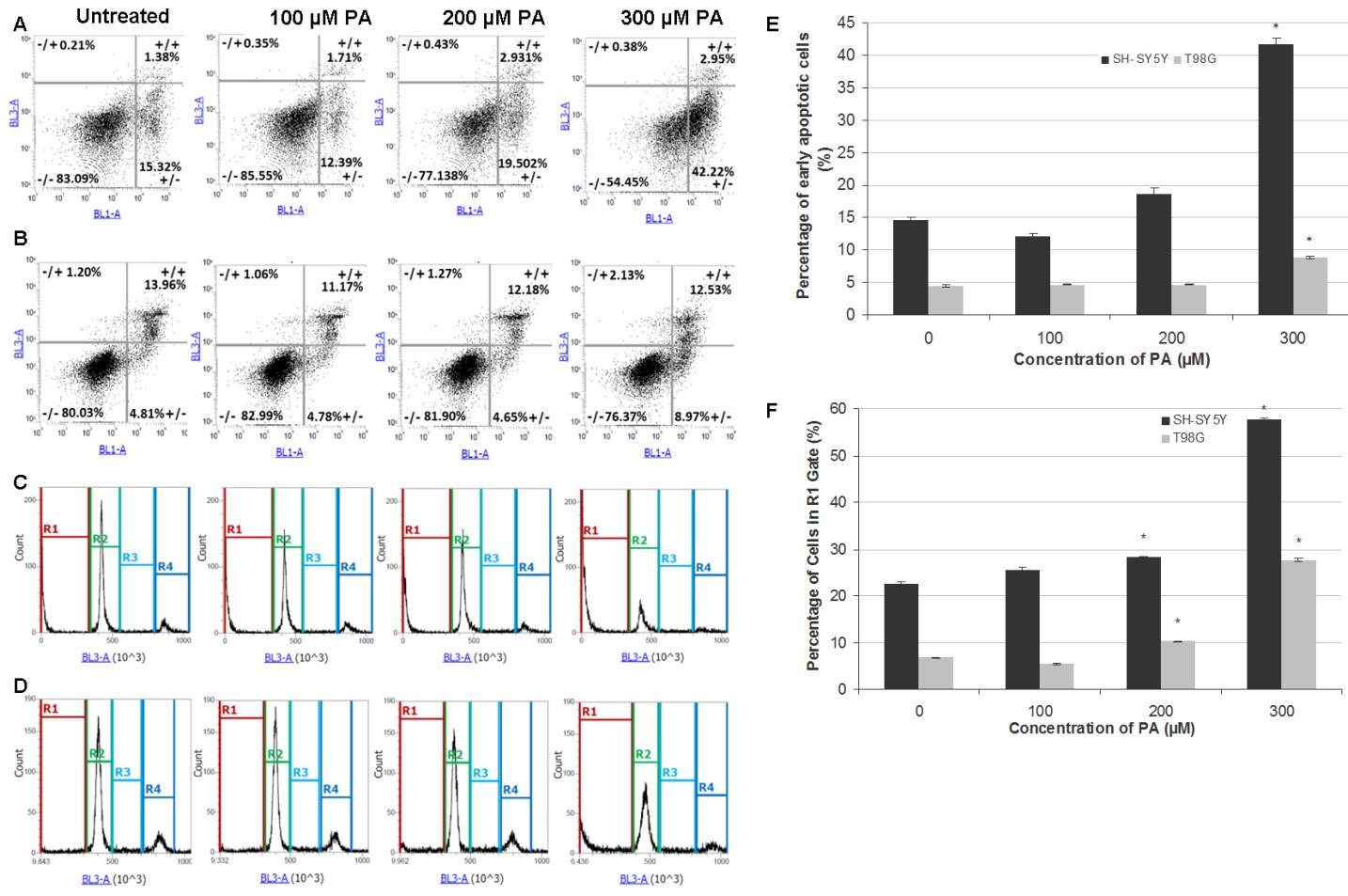

**Figure 3** Annexin V-Alexa Fluor® 488/PI flow cytometric analysis of apoptosis and cell cycle in SH-SY5Y and T98G cells after 48 h of PA treatment. SH-SY5Y and T98G cells were treated with increasing concentrations (0, 100, 200 and 300 $\mu$M) of PA for 48 h. Cells were stained with annexin V-Alexa Fluor® 488 and PI followed by flow cytometric analysis. Representative dot plots from one experiment are shown. $-/+$, necrosis; $+/+$, late apoptosis; $-/-$, live cells and $+/-$, early apoptosis. (A) SH-SY5Y. (B) T98G. Cell cycle distribution was also analysed by flow cytometry. Representative DNA content histograms from one experiment were presented. R1, Sub-G-0/$G_1$ phase; R2, $G_0$/G-1 phase; R3, S phase and R4, $G_2$/M phase. (C) SH-SY5Y. (D) T98G. (E) Statistical graph of the '$+/-$ quadrant' of the dot plots of SH-SY5Y and T98G cells. Data represent the mean $\pm$ S.E.M. of three independent experiments. * represents $p < 0.01$ as compared to untreated. (F) Statistical graph of R1 gate of SH-SY5Y and T98G cells. Data represent the mean $\pm$ S.E.M. of three independent experiments. * represents $p < 0.01$ as compared to untreated.

## PA induces neurotoxicity and gliatoxicity via apoptotic cell death

To investigate the mode of cell death induced by PA in SH-SH5Y and T98G cells, dual staining with Annexin V and PI was performed and cells were analysed by flow cytometry. The results showed that PA at lower concentrations did not induce apoptosis in SH-SY5Y cells (less than 20% cells undergoing apoptosis) and T98G cells (less than 5% cells undergoing apoptosis), but at 300 $\mu$M, PA significantly increased the percentage of apoptotic cells to about 2 fold in both cell lines (Figs. 3A, 3B and 3E). Single staining with PI for cell cycle analysis by flow cytometry also revealed that DNA fragmentation was statistically significant in 200 $\mu$M and 300 $\mu$M PA-treated SH-SY5Y and T98G cells (Figs. 3C, 3D and 3F). For SH-SY5Y cells, the increment in the percentage of DNA
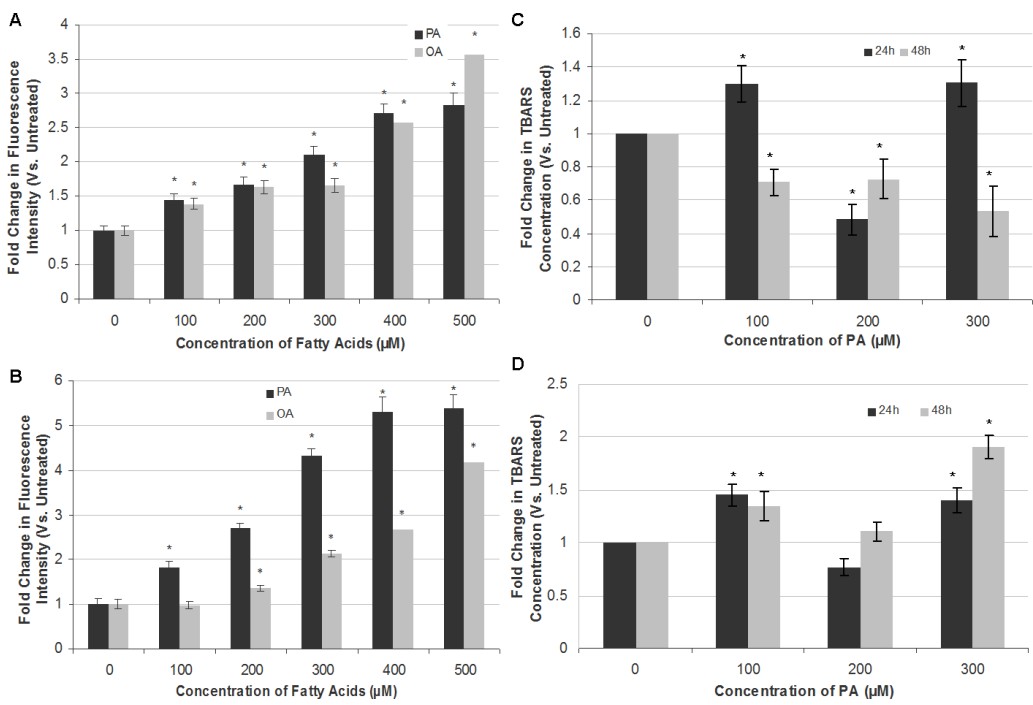

**Figure 4 Measurement of ROS generation and lipid peroxidation in SH-SY5Y and T98G cells after PA and OA treatment.** SH-SY5Y and T98G cells were treated with increasing concentrations (0, 100, 200, 300, 400 and 500 μM) of PA or OA for 6 h and the ROS generation was measured using the DCFH-DA assay. Figure shows the fold change in fluorescent intensity as compared to the untreated group. (A) SH-SY5Y. (B) T98G; Data represent the mean ± S.E.M. of three independent experiments; * represents $p < 0.05$ as compared to the untreated. SH-SY5Y and T98G cells were also treated with increasing concentrations (0, 100, 200 and 300 μM) of PA for 24 h or 48 h and TBARS assay was then performed. (C) SH-SY5Y. (D) T98G. Data represent the mean ± S.E.M. of three independent experiments; * represents $p < 0.05$ as compared to untreated.

fragmentation as indicated by the R1 gate was 1.3-fold (22.56 ± 0.53% to 28.23 ± 0.03%) and 2.6-fold (from 22.56 ± 0.53% to 57.75 ± 0.19%) for 200 μM and 300 μM PA treatments, respectively (Fig. 3F). While for T98G, the increment was 1.5-fold (from 6.74 ± 0.17% to 10.26 ± 0.22%) and 4.1-fold (from 6.74 ± 0.17% to 27.66 ± 0.35%) for 200 μM and 300 μM PA treatments, respectively (Fig. 3F). These were associated with the decrease in cell percentages in the other cell cycle phases.

## PA induces apoptotic cell death in SH-SY5Y and T98G cells via oxidative stress

To attribute whether apoptotic cell death induction by PA in SH-SY5Y and T98G is caused by oxidative stress, intracellular ROS ($H_2O_2$) and extent of lipid peroxidation (TBARS level) were assessed. PA and OA treatments both increased ROS levels in a dose-dependent manner in both SH-SY5Y and T98G cells, with PA inducing greater ROS generation as compared to OA (Fig. 4). PA-induced ROS production was at greater extent in T98G (5.39 ± 0.31-fold) as compared to that in SH-SY5Y (2.83 ± 0.16-fold) at 500 μM PA treatment. OA also elevated the level of ROS in both cell lines but to a degree lesser than PA, except

500 μM OA in SH-SY5Y. However, the changes in the lipid peroxidation level were not dose- and time-dependent in both cell lines, as shown in Fig. 4. In SH-SY5Y, the level of lipid peroxidation increased 0.3-fold to 1.3-fold at 24 h at 100 and 300 μM PA treatments. At 200 μM, the fold change decreased to 0.5-fold as compared to the untreated. For the 48 h treatment of PA, the level of lipid peroxidation decreased gradually to 0.7-fold of the untreated at 100 and 200 μM, and to 0.5-fold at 300 μM. In T98G, 24 h treatment of PA increased the level of lipid peroxidation to 1.5- and 1.4-fold at 100 and 300 μM, respectively, as compared to the untreated. 200 μM PA decreased the level to 0.8-fold. At 48 h, PA treatment on the cell line increased the lipid peroxidation level to 1.3-, 1.1- and 1.8-fold of that of the untreated at 100, 200 and 300 μM, respectively. In short, 300 μM of PA treatment impacted the lipid peroxidation level the most. The changes of lipid peroxidation level in SH-SY5Y were lesser as compared to that of T98G, in which the latter fold change was increased to 1.9-fold at 300 μM after 48 h of PA treatment.

## DISCUSSION

MTT assay results revealed that only PA, but not OA and LA, is cytotoxic to all of the three cell lines; namely, SH-SY5Y, SH-SY5Y- γ and T98G. The range of FA concentrations used was within the physiological range of circulating free FAs in human plasma, which is ~0.1–1.0 mM (*Yuan et al., 2013*). The rate of PA β-oxidation was also found to increase as a function of PA concentration, from 15 μM to 2 mM, and peaked at 200 μM, in homogenates of astrocytes cultured from neonatal mouse brain (*Murphy et al., 1992*). The similar range of FA concentrations had been used in many studies, and the cytotoxic effect of PA observed was also similar to that reported earlier (*Gupta et al., 2012*; *Yuan et al., 2013*; *Hsiao et al., 2014*). Dose- and time-dependent apoptotic effects were observed in SH-SY5Y after PA treatment (*Kim et al., 2011*). Further elucidation of pathways leading to PA-induced apoptosis revealed that endoplasmic reticulum (ER) stress, as indicated by the expression of spliced X-box binding protein 1 mRNA and immunoglobin heavy chain-binding protein, was involved (*Kim et al., 2011*). In addition, PA inactivated 5′ adenosine monophosphate-activated protein kinase (AMPK), and re-activation of AMPK by N1-(β-D-Ribofuranosyl)-5-aminoimidazole-4-carboxamide, ameliorated PA-induced cytotoxicity with diminished ER stress-mediated apoptosis (*Kim et al., 2011*). Correspondingly, *Hsiao et al. (2014)* also showed that PA triggered SH-SY5Y apoptotic cell death via protein palmitoylation, which induced $G_2/M$ cell cycle arrest and ER stress.

On the other hand, OA was shown to promote the cell growth in both neuronal SH-SY5Y and astrocytic T98G cell lines, parallel to previous studies in which OA was found to have neuroprotective and neurotrophic effects. For example, OA was found to be superior to docosahexaenoic acid or linoleic acid in the protection of mouse neuroblastoma Neuro-2a and primary rat cortical neurons against PA- or ceramide-induced cytotoxicity (*Kwon, Lee & Querfurth, 2014*). OA pre-treatment attenuated PA-induced mitochondrial dysfunction and insulin resistance by inhibiting the phosphorylation of mitogen-activated protein kinase and nuclear translocation of NF-κB p65 induced by PA (*Kwon, Lee & Querfurth, 2014*). It also stimulated adenosine triphosphate (ATP) production, decreased mitochondrial

superoxide generation, increased triglyceride and blocked PA-induced diacylglycerol accumulation, with the involvement of protein kinase A and peroxisome proliferator-activated receptor gamma coactivator-1α (*Kwon, Lee & Querfurth, 2014*). In addition to its neuroprotective effect, OA was found to be synthesised and released by astrocytes with the stimulation of albumin (*Tabernero et al., 2001*). Released OA was then incorporated into neurons for the synthesis of phospholipids, and stimulated neuronal differentiation by promoting axonal growth, neuronal clustering and expression of axonal growth-associated protein-43 (*Tabernero et al., 2001*). The neurotrophic effect of OA involved the binding to its receptor peroxisome proliferator-activated receptor-α (*Bento-Abreu, Tabernero & Medina, 2007*).

The same cytotoxic effect was also observed in the astrocytic cell line, T98G. This PA-induced cytotoxicity was also reported previously in primary cultured mouse astrocytes (*Wang et al., 2012*) and rat cortical astrocytes (*Wong et al., 2014*). *Wang et al. (2012)* showed that PA-induced apoptosis in primary mouse astrocytes involved the rise in Bax/Bcl-2 ratio and caspase-3 activation. It involved ROS generation and subsequent mitochondrial membrane potential collapse (*Wang et al., 2012*; *Wong et al., 2014*). PA was also found to induce inflammatory response in astrocytes via Toll-like receptor 4, by releasing pro-inflammatory cytokines like tumour necrosis factor-α and interleukin-6 (*Gupta et al., 2012*). The inflammatory response was also observed with the treatment of other SFAs, namely, LA and stearic acid, but not unsaturated FAs (like OA) (*Gupta et al., 2012*).

PA induced cytotoxicity in both SH-SY5Y and SH-SY5Y-γ, albeit to a lesser extent in the latter, as indicated by the higher $LC_{50}$. As a marker of cancer progression, overexpression of γ-syn in ovarian cancer cell lines, A2780 and OVCAR5, was found to enhance tumorigenicity by constitutively activating Extracellular Signal-regulated Kinase-1/2 and down-regulating c-Jun N-terminal kinase 1 in response to a host of environmental stress signals (*Pan et al., 2002*). In human breast cancer cell lines T47D and SKBR3, knockdown of γ-syn sensitized the cells to ER stress-induced apoptosis dependent on c-Jun N-terminal kinases or caspase-3 and caspase-7 activation (*Hua et al., 2009*). Consistent with these, our study suggests an anti-apoptotic role of γ-syn in response to PA-induced cytotoxicity, since the two cells lines used are cancer cell lines.

α-syn and γ-syn, except A53T α-syn, generally increased the cell viability of both SH-SY5Y and T98G cells after PA treatment. Previously, *Da Costa, Ancolio & Checler (2000)* reported that wt α-syn, but not A53T, had anti-apoptotic effect against staurosporine, etoposide and ceramide C in murine neuronal cell line TSM1. Similarly, our group reported that only wt α-syn, but not the mutant variants, protected SH-SY5Y from the toxicity of rotenone and maneb (Parkinsonian pesticides), by attenuating mitochondrial membrane potential changes and ROS level (*Choong & Say, 2011*). On the other hand, overexpression of α-syn led to oxidative stress-mediated apoptotic death in U373 astrocytoma cells (*Stefanova et al., 2001*). In this study, we found that transient overexpression of not only wt, but also A30P, E46K α-syn and γ-syn, were able to ameliorate (albeit modestly) PA-induced cytotoxicity in SH-SY5Y and T98G cells. The discrepancy in the cytotoxic effect of α-syn overexpression might be due to the transfection method used (transient

in this study *vs.* stable in *Stefanova et al., 2001*). A further confirmatory study involving stable overexpression of α-syn and its PD mutants in SH-SY5Y and T98G cells is needed to further elucidate whether α-syn protects against PA-induced cytotoxicity, since α-syn has been shown to bind to the surface of triglyceride-rich lipid droplets and protects them from hydrolysis in cultured cells as well as in primary neurons (*Cole et al., 2002*).

The co-treatment of PA and PQ brought about intensified cytotoxicity in SH-SY5Y cells as compared to PA or PQ treatment alone. There is no previous study reporting on the interaction between PA and PQ in dopaminergic neurotoxicity, but rotenone, another Parkinsonian environmental toxin which inhibits mitochondrial complex I, was found to increase the incorporation of radioactively-labeled PA into acetyl coA by β-oxidation in SH-SY5Y cells (*Worth et al., 2014*). In bovine cerebral mitochondria, PQ was reduced to the PQ radical via complex I in mitochondria, leading to accelerated lipid peroxidation, an effect similar to that triggered by rotenone (*Fukushima et al., 1995*). It is predicted that the PQ-induced lipid peroxidation is further enhanced by PA, which serves as a substrate for carnitine palmitoyltransferase-1-dependent mitochondrial β-oxidation, a process that leads to enhanced ROS production (*Magtanong, Ko & Dixon, 2016*). In addition, PA treatment also triggers mitogen-activated protein kinase- and caspase-dependent signaling pathways leading to apoptosis in neuronal N2a cells (*Kwon, Lee & Querfurth, 2014*). PQ was found to trigger SH-SY5Y neuronal cell apoptosis via the intrinsic mitochondrial pathway by the upregulation of p53 protein and subsequently, its target pro-apoptotic Bax protein (*Yang & Tiffany-Castiglioni, 2008*). The impairment of mitochondria complex I activity was followed by the release of cytochrome *c*, increased caspases 9 and 3 activities, nuclear condensation and DNA fragmentation (*Yang & Tiffany-Castiglioni, 2008*). Thus, it is suggested that PQ and PA synergistically enhance SH-SY5Y cytotoxicity.

Leptin, secreted primarily by adipocytes, is transported across the blood–brain barrier and acts on leptin receptors in CNS to regulate food intake by modulating activity of appetite control neurons in the brain (*Zhang et al., 1994*). Obesity is associated with leptin resistance, where high plasma leptin concentration was observed in most obese humans and rodents but this hyperleptinemia may not reduce appetite or increase energy expenditure (*Frederich et al., 1995*; *Considine et al., 1996*). It was reported that leptin receptors, the long and the short isoforms, are expressed in SH-SY5Y (*Russo et al., 2004*). However, in this study, pre-treatment of leptin was found to exert no neuroprotective effect against PA-induced cell death in SH-SY5Y cells. In a previous study, leptin was found to stimulate cell proliferation in a dose- and time-dependent manner involving the mechanism of apoptosis suppression in SH-SY5Y cells (*Russo et al., 2004*). SFAs (including PA) were shown to induce ER stress and inflammatory response via toll-like receptor 4 signalling, leading to leptin resistance in rat hypothalamus, whereas rats fed with monounsaturated FAs (including OA) did not develop hypothalamic leptin resistance (*Milanski et al., 2009*). Therefore, the result suggests a PA-induced leptin resistance in neurons, diminishing its neuroprotective effect.

The apoptotic cell death was confirmed with flow cytometric analysis using annexin V/PI staining and cell cycle analysis. The results are consistent with previous studies (*Wang et al., 2012*; *Hsiao et al., 2014*). In SH-SY5Y, PA induced neuron cell cycle $G_2$/M arrest at

24 h and increased in sub-$G_0$ phase at 48 h (*Hsiao et al., 2014*). However, the percentage of apoptotic cells in T98G as quantified by flow cytometry was much lower than that in SH-SY5Y, despite the similar $LC_{50}$ obtained from the MTT assay. This indicates that the use of MTT assay has a limitation, as its rate of tetrazolium reduction may reflect the general metabolic activity or the rate of glycolytic NADH production, and not the cell number (*Berridge, Herst & Tan, 2005*). The lesser PA-induced apoptotic cells observed in astrocytic T98G as compared to neuronal SH-SY5Y cells was not in the odd, owing to the high FA β-oxidation rate in astrocytes as compared to the poor utilisation of FA as fuel in neurons, as discussed earlier (*Schönfeld & Reiser, 2013*).

In both cell lines, PA and OA treatments were shown to induce ROS generation in a dose-dependent manner. However, the increase in ROS level was found to be higher in T98G than in SH-SY5Y. Also, the degree of lipid peroxidation was higher in T98G cells as compared to that in SH-SY5Y cells. However, there is a similar pattern of increment at 100 µM PA, decrement at 200 µM and rise again at 300 µM in both cell lines, except for the 48 h treatment in SH-SY5Y. At 48 h, the fold change in lipid peroxidation degree gradually decreased with increasing PA concentrations in SH-SY5Y. These observations may be attributed by the high FA β-oxidation rate in astrocytes, a prominent source of ROS generation (*Seifert et al., 2010*; *Rodrigues & Gomes, 2012*). β-oxidation of PA yields 15 molecules of flavin adenine dinucleotide ($FADH_2$) and 31 molecules of nicotinamide adenine dinucleotide (NADH) ($FADH_2$/NADH ratio ≈ 0.5) as compared to that of glucose degradation in which 2 molecules of $FADH_2$ and 10 molecules of NADH are generated ($FADH_2$/NADH ratio = 0.2) (*Schönfeld & Reiser, 2013*). Enhanced ROS generation is observed during the oxidation of $FADH_2$ for ATP generation by the electron transfer flavoprotein-ubiquinone oxidoreductase. Thus, the higher $FADH_2$/NADH ratio of PA β-oxidation in astrocytes would result in elevated ROS level. On the other hand, OA treatment in both cell lines was also shown to increase the ROS level, despite no cytotoxicity effect was detected by MTT assay. This is not unexpected, as OA was found to induce the production of ROS in rat aortic smooth muscle cells (*Lu et al., 1998*), human neutrophil (*Carrillo et al., 2011*) and cultured endothelial cells (bEnd.3) (*Gremmels et al., 2015*).

## CONCLUSIONS

In summary, PA, but not OA and LA, induced cytotoxicity in SH-SY5Y, SH-SY5Y-γ and T98G cell lines in a time- and dose-dependent manner. The PA-induced cytotoxicity was found to be lower in SH-SY5Y-γ, suggesting its possible role in neuroprotection. Co-treatment of PA and PQ revealed that the PA-induced cytotoxicity was exacerbated by PQ. Leptin did not protect SH-SY5Y cell line from PA-induced neurotoxicity, suggesting a PA-induced leptin resistance. Annexin V/PI and sub-$G_0$ cell cycle analysis by flow cytometry revealed that PA-induced apoptotic cell death in both SH-SY5Y and T98G cell lines, but the percentage of apoptosis was much lower in T98G with similar concentrations of PA treatment. This indicates that neurons are more susceptible to PA-induced cytotoxicity. The PA-induced apoptotic cell death was found to be associated with increased lipid peroxidation and ROS generation. Taken together, the results suggest that HFD may cause

neuronal and astrocytic damage, which indirectly proposes that CNS pathologies involving neuroinflammation and reactive gliosis could be prevented via the diet regimen. Apart from that, the signalling pathways involved in PA-induced apoptotic cell death and the neuroprotection of γ-syn warrant further investigation.

### Funding
This work was supported by the Malaysian Ministry of Higher Education Fundamental Research Grant Scheme (FRGS/1/2013/SKK01/UTAR/02/1). The funders had no role in study design, data collection and analysis, decision to publish, or preparation of the manuscript.

### Grant Disclosures
The following grant information was disclosed by the authors:
Malaysian Ministry of Higher Education Fundamental Research Grant Scheme: FRGS/1/2013/SKK01/UTAR/02/1.

### Competing Interests
The authors declare there are no competing interests.

### Author Contributions
- Yee-Wen Ng conceived and designed the experiments, performed the experiments, analyzed the data, contributed reagents/materials/analysis tools, prepared figures and/or tables, authored or reviewed drafts of the paper, approved the final draft, performed statistical analysis.
- Yee-How Say analyzed the data, contributed reagents/materials/analysis tools, prepared figures and/or tables, authored or reviewed drafts of the paper, approved the final draft, performed statistical analysis.

### Data Availability
Say, Yee-How (2018): PA SHT98 Paper Raw Data YHSay. figshare. Fileset. DOI 10.6084/m9.figshare.5924110.v6.

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
