# Peer review of "Palmitic acid induces neurotoxicity and gliatoxicity in SH-SY5Y human neuroblastoma and T98G human glioblastoma cells"

_PeerJ, doi:10.7717/peerj.4696_

## Round 0.1 · original submission · Minor Revisions

Dear Dr. Yee-How Say,

Thank you for submitting your paper to PeerJ. It now been reviewed and I have the following recommendations in line with the reviewers' reports.

The study contributes to our growing knowledge of the negative influence of high-fat diet on the CNS. The conclusion that palmitic acid induces apoptosis by increasing oxidative stress in neurons and astrocytes is interesting. The paper, while interesting, should be modified to address several points prior to publication.

The reviewers provided detailed comments, and I ask that you consider these carefully and correct numerous errors when revising the manuscript. I would like to pay attention to the following comments:
One of the problems is the experimental design. You should describe the experiments in details and give reasons for choosing the treatment with 300 μM PA, OA or LA for 24 for ORO staining as well as to justify the choice of treatment time as 48 h for apoptosis detection. Additionally, you should clarify the conditions of cytometrical analysis and give the additional information on minimal quantity of analysed events in samples and the order of analysis of obtained cytometrical data.

The reviewer recommends you discussed the increasing viability of cells at low concentrations of PA, OA and LA. What can be molecular mechanisms of this phenomenon? It may be interesting.

The resolution of the Figure 3 (ORO staining of SH-SY5Y and T98G after fatty acid treatments) in insufficient, it should be improved.

English should be improved upon before acceptance.

I look forward to receiving your revision.

·

Basic reporting

The structure is acceptable, but the titles of some paragraphs and figures demand re-considering and re-naming. The manuscript contains too many unreasonable abbreviations. Cytometrical data (FCS files) are not presented in Raw data.

Experimental design

Cytometrical analysis needs elucidation: additional information on minimal quantity of analysed events in samples and the order of analysis of obtained cytometrical data.

Validity of the findings

Microscopic photos and cytometrical figures demand improvement.

Additional comments

The manuscript is interesting and necessary for the biomedical science in general, but needs major revision for improvement. The results obtained are believable, important and well discussed, but need some corrections in their presenting. Please, find my comments in attached PDF file.

Reviewer 2 ·

Basic reporting

The manuscript contains many typographical, punctuation and other errors. Also English is not clear, unambiguous and professional throughout the manuscript. Please be more conscientious. Some examples of those errors:

1. When authors use abbreviation for the first time, they must give the full term (in line 27 for CNS, in line 85 for BMI).
2. Scale bars are absent in Figure 2. Moreover, it seems that pictures of T98G untreated and 300 μM PA treated cells were made with using different magnification.
3. Full stop is absent in line 474 in the end of sentence.
4. In Figure 1 is not clear which one of the diagrams describe 24 h and 48 h of treatment.
5. In some points authors give unnecessary information (in line 231it is enough to write just final concentration).
6. In Figure 4 E what does percentage of cells means? It is not clear from diagram.

Experimental design

Research question of this manuscript is well defined and relevant, but methods described with sufficient detail not in all cases. Also there are some concerns on experimental design.

Why authors chose treatment for 48 h for apoptosis detection? To my knowledge is too long. Besides, did they do all necessary controls for these experiments?

Also it is not clear why authors chose 300 μM treatment with 300 μM PA, OA or LA for 24 for ORO staining.

Authors did not give concentration of paraformaldehyde (Line 195) and model of camera. Please check is duration of fixation that you give is real (it seems too long).

Authors did not include Western blot for γ-synuclein protein expression after establishment of stable cells in the manuscript, but I expect to see it.

Validity of the findings

Data is mostly robust and statistically sound, but appropriate controls for apoptosis detection are absent.

Authors did not discussed in details why PA, OA and LA can increase viability of cells. What can be molecular mechanisms of this?

---

## Round 0.2 · accepted · Accept

Dear Yee-How Say,

Thank you for submitting the revised version of your manuscript to PeerJ.

In the revised version the authors took into consideration all comments and remarks. I recommend to accept your manuscript for publication in PeerJ.

#